Journal of Data-centric Machine Learning Research (2024)          Submitted 01/24; Revised 07/24; Published 08/24

# FedAIoT: A Federated Learning Benchmark for Artificial Intelligence of Things

**Samiul Alam**[1]                                                                                    ALAM.140@OSU.EDU
**Tuo Zhang**[2]                                                                                      TUOZHANG@USC.EDU
**Tiantian Feng**[2]                                                                                  TIANTIAF@USC.EDU
**Hui Shen**[1]                                                                                       SHEN.1780@OSU.EDU
**Zhichao Cao**[3]                                                                                    CAOZC@MSU.EDU
**Dong Zhao**[3]                                                                                      DZ@MSU.EDU
**JeongGil Ko**[4]                                                                                    JEONGGIL.KO@YONSEI.AC.KR
**Kiran Somasundaram**[5]                                                                             KIRANSOM@META.COM
**Shrikanth S. Narayanan**[2]                                                                         SHRI@SIPI.USC.EDU
**Salman Avestimehr**[2]                                                                              AVESTIME@USC.EDU
**Mi Zhang**[1]                                                                                       MIZHANG.1@OSU.EDU

[1] *The Ohio State University*
[2] *University of Southern California*
[3] *Michigan State University*
[4] *Yonsei University*
[5] *Meta*

**Reviewed on OpenReview:** *https://openreview.net/forum?id=fYNw9Ukljz*

**Editor:** Peter Mattson

## Abstract

There is a significant relevance of federated learning (FL) in the realm of Artificial Intelligence of Things (AIoT). However, most of existing FL works are not conducted on datasets collected from authentic IoT devices that capture unique modalities and inherent challenges of IoT data. To fill this critical gap, in this work, we introduce `FedAIoT`, an FL benchmark for AIoT. `FedAIoT` includes eight datasets collected from a wide range of IoT devices. These datasets cover unique IoT modalities and target representative applications of AIoT. `FedAIoT` also includes a unified end-to-end FL framework for AIoT that simplifies benchmarking the performance of the datasets. Our benchmark results shed light on the opportunities and challenges of FL for AIoT. We hope `FedAIoT` could serve as an invaluable resource to foster advancements in the important field of FL for AIoT. The repository of `FedAIoT` is maintained at https://github.com/AIoT-MLSys-Lab/FedAIoT.

**Keywords:** Benchmarks, Federated Learning, Artificial Intelligence of Things, AIoT

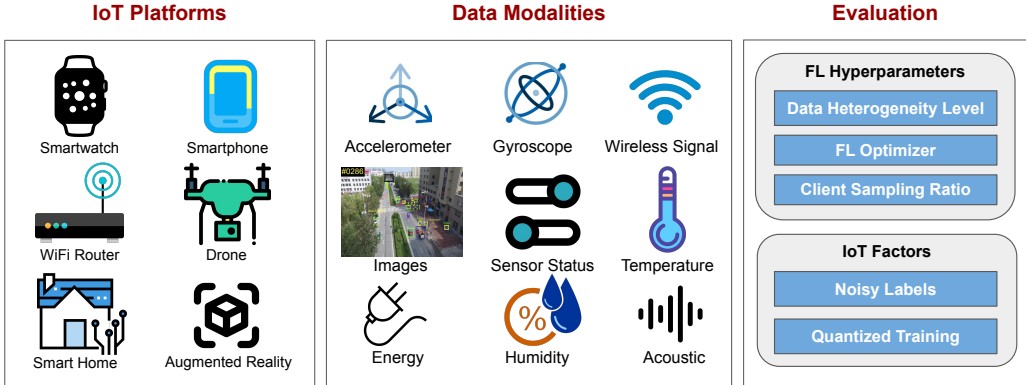

Figure 1: Overview of `FedAIoT`.

# 1 Introduction

Internet of Things (IoT) such as smartphones, drones, and sensors deployed at homes are ubiquitous today. The advances in Artificial Intelligence (AI) have boosted the integration of IoT and AI that turns Artificial Intelligence of Things (AIoT) into reality. However, data captured by IoT devices usually contain privacy-sensitive information. In recent years, federated learning (FL) has emerged as a privacy-preserving solution that allows the extraction of knowledge from collected data while keeping the data solely on the devices (Kairouz et al., 2021; Wang et al., 2021; Zhang et al., 2022).

Despite the significant relevance of FL in the AIoT realm, as summarized in Table 9 in Appendix A, most existing FL works are conducted on well-known datasets such as CIFAR-10 and CIFAR-100. *These datasets, however, do not originate from authentic IoT devices and thus fail to capture the unique modalities and inherent challenges associated with real-world IoT data.* This discrepancy underscores a strong need for an IoT-oriented FL benchmark to fill this critical gap.

In this work, we present `FedAIoT`, an FL benchmark for AIoT (Figure 1). At its core, `FedAIoT` includes eight well-chosen datasets collected from a wide range of IoT devices from smartphones, smartwatches, and Wi-Fi routers, to drones, smart home sensors, and head-mounted devices that either have already become an indispensable part of people's daily lives or are driving emerging applications. These datasets encapsulate a variety of unique IoT-specific data modalities such as wireless data, drone images, and smart home sensor data (e.g., motion, energy, humidity, temperature) that have not been explored in existing FL benchmarks. Moreover, these datasets target some of the most representative applications and innovative use cases of AIoT.

To facilitate benchmarking and ensure reproducibility, `FedAIoT` includes a unified end-to-end FL framework for AIoT, which covers the complete FL-for-AIoT pipeline: from non-independent and identically distributed (non-IID) data partitioning, IoT-specific data preprocessing, to IoT-friendly models, FL hyperparameters, and IoT-factor emulator. Our framework also includes the implementations of popular schemes, models, and techniques involved in each stage of the FL-for-AIoT pipeline.

We have conducted systematic benchmarking on the eight datasets using the end-to-end framework. Specifically, we examine the impact of varying degrees of non-IID data distributions, FL optimizers, and client sampling ratios on the performance of FL. We also

Table 1: Comparison between `FedAIoT` and existing FL benchmarks.

| | Data Type | FL Framework Designed for IoT | Noisy Labels | Quantized Training |
|---|---|---|---|---|
| FLamby | Medical Images | No | No | No |
| FedAudio | Audio Data | No | Uniform | No |
| FedMultimodal | Multimodality Data | No | Uniform | No |
| FedGraphNN | Graph Data | No | No | No |
| FedNLP | Natural Language | No | No | No |
| FLUTE | Images and Text | No | No | Server Side only |
| FedCV | Images | No | No | No |
| **FedAIoT** | **IoT Data** | **Yes** | **Probabilistic** | **Server and Client Side** |

evaluate the impact of noisy labels, a prevalent challenge in IoT datasets, as well as the effects of quantized training, a technique that tackles the practical limitation of resource-constrained IoT devices. Our benchmark results provide valuable information about both the opportunities and challenges of FL for AIoT. Given the significant relevance of FL in the realm of AIoT, we hope `FedAIoT` could act as a valuable tool to promote advancements in the important area of FL for AIoT. In summary, our work makes the following contributions.

- **IoT-focused Benchmark.** Our benchmark represents the first FL benchmark that focuses on data collected from a diverse set of authentic IoT devices. Moreover, our benchmark includes unique IoT-specific data modalities that previous benchmarks do not include.

- **Unified End-to-End FL Framework for AIoT.** We introduce the first unified end-to-end FL framework for AIoT that covers the complete FL-for-AIoT pipeline: from non-IID data partitioning, and IoT-specific data preprocessing, to IoT-friendly models, FL hyperparameters, and IoT-factor emulators.

- **Novel Design of Noisy Labels.** Our benchmark also introduces a novel way to design noisy labels. Real-world FL deployments on IoT devices often encounter data labeling errors, which act as noises in federated training. To emulate such noises, different from prior benchmarks that adopt either uniform error distribution or non-uniform error distributions with the assumption that one label can only be mislabeled as another specific label with a random probability, we have designed a new label transition probability matrix based on the insight that labels that are similar to each other are more likely to be mislabeled. This design allows us to simulate data label noises with more realistic transition probabilities and to benchmark FL algorithms for their robustness to such noises.

- **Quantized Training.** Lastly, we are also the first FL benchmark to show the effect of quantized training on both server and client sides in the context of FL. Previous benchmarks such as FLUTE (Garcia et al., 2022) demonstrate the effect of quantized training during server-side aggregation for communication reduction. In contrast, our benchmark also incorporates quantized training during client-side training to reduce the memory demands of FL on the device. This is a key difference as IoT devices are often limited by not onlycommunication bandwidth but also on-device memory capacity.

Table 2: Overview of the datasets included in `FedAIoT`.

| Dataset | IoT Platform | Data Modality | Data Dimension | Dataset Size | # Training Samples | # Clients |
|---|---|---|---|---|---|---|
| WISDM-W | Smartwatch | Accelerometer Gyroscope | $200 \times 6$ | 294 MB | $16,569$ | 80 |
| WISDM-P | Smartphone | Accelerometer Gyroscope | $200 \times 6$ | 253 MB | $13,714$ | 80 |
| UT-HAR | Wi-Fi Router | Wireless Signal | $3 \times 30 \times 250$ | 854 MB | $3,977$ | 20 |
| Widar | Wi-Fi Router | Wireless Signal | $22 \times 20 \times 20$ | 3.3 GB | $11,372$ | 40 |
| VisDrone | Drone | Images | $3 \times 224 \times 224$ | 1.8 GB | $6,471$ | 30 |
| CASAS | Smart Home | Motion Sensor Door Sensor Thermostat | $2000 \times 1$ | 233 MB | $12,190$ | 60 |
| AEP | Smart Home | Energy, Humidity Temperature | $18 \times 1$ | 12 MB | $15,788$ | 80 |
| EPIC-SOUNDS | Augmented Reality | Acoustics | $400 \times 128$ | 34 GB | $60,055$ | 210 |

## 2 Related Work

The importance of data to FL research pushes the development of FL benchmarks on a variety of data modalities. Existing FL benchmarks, however, predominantly center around curating FL datasets in the domain of computer vision (CV) (He et al., 2021b; Garcia et al., 2022), natural language processing (NLP) Lin et al. (2022); Garcia et al. (2022), medical imaging (du Terrail et al., 2022), speech and audio (Zhang et al., 2023; Garcia et al., 2022), and graph neural networks (He et al., 2021a). For example, FedCV He et al. (2021b), FedNLP (Lin et al., 2022), and FedAudio (Zhang et al., 2023) focuses on benchmarking CV, NLP, and audio-related datasets and tasks respectively; FLUTE (Garcia et al., 2022) covers a mix of datasets from CV, NLP, and audio; FLamby (du Terrail et al., 2022) primarily focuses on medical images; and FedMultimodal (Feng et al., 2023) includes multimodal datasets in the domain of emotion recognition, healthcare, multimedia, and social media. As summarized in Table 1, although these benchmarks have contributed to FL research, a dedicated FL benchmark explicitly tailored for IoT data is absent. Compared to these existing FL benchmarks, `FedAIoT` is specifically designed to fill this critical gap by providing a dedicated FL benchmark on data collected from a wide range of authentic IoT devices.

## 3 Design of FedAIoT

### 3.1 Datasets

**Objective:** The objective of `FedAIoT` is to provide a benchmark consisting of well-validated and high-quality datasets collected from a wide range of IoT devices, sensor modalities, and applications. *We have made significant efforts to examine a much larger pool of existing datasets and select the high-quality ones to include in our benchmark.* Table 2 provides an overview of the eight high-quality datasets included in `FedAIoT`. These datasets have diverse sizes (small: less than 7k samples; medium: 11k to 16k samples; and large: more than 60k samples). The rationale behind this design choice is to accommodate researchers with different computing resources. For example, researchers with limited computing resources can still pick the relatively small datasets in our benchmark to develop and evaluate their

algorithms. In this section, we provide a brief overview of each included dataset. More details about the datasets are provided in Appendix B and Table 10.

**WISDM:** The Wireless Sensor Data Mining (WISDM) dataset (Weiss et al., 2019; Lockhart et al., 2011) is one of the widely used datasets for the task of daily activity recognition using accelerometer and gyroscope sensor data collected from smartphones and smartwatches. WISDM includes data collected from 51 subjects performing 18 daily activities, each in a 3-minute session. We combined activities such as eating soup, chips, pasta, and sandwiches into a single category called "eating", and removed uncommon activities related to playing with balls, such as kicking, catching, or dribbling. We randomly selected 45 subjects as the training set and the remaining six subjects were assigned to the test set. Given that the smartwatch data and smartphone data were not collected simultaneously for most subjects and thus were not synchronized precisely, we partition WISDM into two independent datasets: **WISDM-W** with smartwatch data only and **WISDM-P** with smartphone data only. The total number of samples in the training and test set is $16,569$ and $4,103$ for WISDM-W and $13,714$ and $4,073$ for WISDM-P respectively. No Licence was explicitly mentioned on the dataset homepage.

**UT-HAR:** The UT-HAR dataset (Yousefi et al., 2017) is a dataset for contactless activity recognition based on Wi-Fi signals. The Wi-Fi data are in the form of Channel State Information (CSI) collected using three pairs of antennas and an Intel 5300 Network Interface Card (NIC), with each antenna pair capable of capturing 30 subcarriers of CSI. UT-HAR comprises data collected from subjects performing seven activities such as walking and running. UT-HAR contains a pre-determined training and test set. The total number of training and test samples is $3,977$ and $500$ respectively. No Licence was mentioned on the dataset homepage.

**Widar:** The Widar dataset (Yang, 2020; Zheng et al., 2019) is designed for contactless gesture recognition using Wi-Fi signal strength measurements collected from strategically placed access points. The data collection system uses an Intel 5300 NIC with $3 \times 3$ antenna pairs. The dataset includes data from 17 subjects performing 22 unique gestures like push, pull, sweeping, and clapping. There are some gesture classes that were only contributed by one subject. Including those would risk the model overfitting for that subject instead of the gesture itself. As such, only gestures recorded by more than three subjects are retained, with data from two subjects used during training and the remaining one for the test set. This decision ensures that the test set does not have data from the same subjects as the training set. The resulting dataset includes nine gestures with $11,372$ samples in the training set and $5,222$ in the test set. The dataset is licensed under the Creative Commons Attribution-NonCommercial 4.0 International Licence (CC BY 4).

**VisDrone:** The VisDrone dataset (Zhu et al., 2021) is a large-scale dataset dedicated to object detection in aerial images captured by drone cameras. VisDrone includes a total of 263 video clips, which contain $179,264$ frames and $2,512,357$ labeled objects. The labeled objects fall into 12 categories (e.g., "pedestrian", "bicycle", and "car"), recorded under various scenarios such as crowded urban areas, highways, and parks. The dataset contains a pre-determined training and test set. The total number of samples in the training and test set is $6,471$ and $1,610$ respectively. The dataset is licensed under Creative Commons Attribution-NonCommercial-ShareAlike 3.0 License.

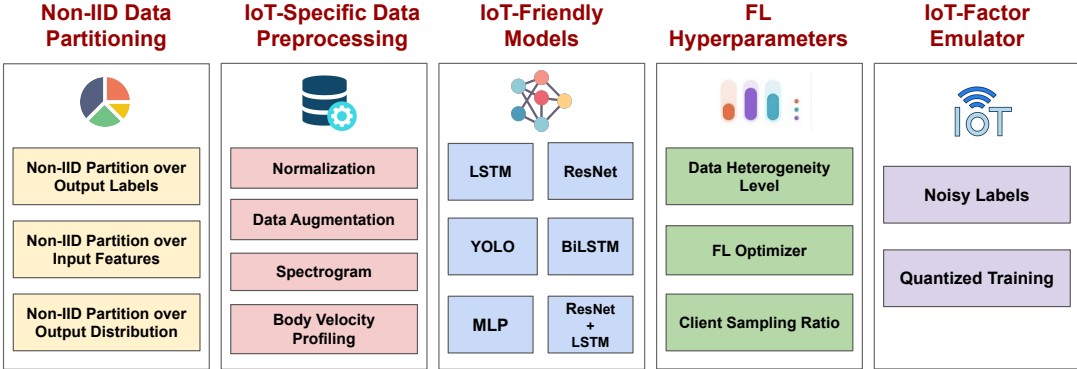

Figure 2: Overview of the end-to-end FL framework for AIoT included in `FedAIoT`.

**CASAS:** The CASAS dataset (Schmitter-Edgecombe and Cook, 2009), derived from the CASAS smart home project, is a smart home sensor dataset for the task of recognizing activities of daily living (ADL) based on sequences of sensor states over time to support the application of independent living. Data were collected from three distinct apartments, each equipped with three types of sensors: motion sensors, temperature sensors, and door sensors. We have selected five specific datasets from CASAS named "Milan", "Cairo", "Kyoto2", "Kyoto3", and "Kyoto4" based on the uniformity of their sensor data representation. The original ADL categories within each dataset have been consolidated into 11 categories related to home activities such as "sleep", "eat", and "bath". Activities not fitting within these categories were collectively classified as "other". The training and test set was made using an 80-20 split. Each data sample is a categorical time series of length 2,000, representing sensor states over a certain period. The total number of samples in the training and test set is 12,190 and 3,048 respectively. No Licence was mentioned on the dataset homepage.

**AEP:** The Appliances Energy Prediction (AEP) dataset (Candanedo et al., 2017) is another smart home sensor dataset but designed for the task of home energy usage prediction. Data were collected from energy sensors, temperature sensors, and humidity sensors installed inside a home every 10 minutes over 4.5 months. The number of samples in the training and test set is 15,788 and 3,947 respectively. No Licence was mentioned on the dataset homepage.

**EPIC-SOUNDS:** The EPIC-SOUNDS dataset (Huh et al., 2023) is a large-scale collection of audio recordings for audio-based human activity recognition for augmented reality applications. The audio data were collected from a head-mounted microphone, containing more than 100,000 categorized segments distributed across 44 distinct classes. The dataset contains a pre-determined training and test set. The total number of training and test samples is 60,055 and 40,175 respectively. The dataset is under CC BY 4 Licence.

### 3.2 End-to-End Federated Learning Framework for AIoT

To benchmark the performance of the datasets and facilitate future research on FL for AIoT, we have designed and developed an end-to-end FL framework for AIoT as another key part of `FedAIoT`. As illustrated in Figure 2, our framework covers the complete FL-for-AIoT pipeline, including (1) non-IID data partitioning, (2) IoT-specific data preprocessing, (3) IoT-friendly

Table 3: Non-IID data partitioning schemes and models used for each dataset.

| Dataset | WISDM-W | WISDM-P | UT-HAR | Widar | VisDrone | CASAS | AEP | EPIC-SOUNDS |
|---|---|---|---|---|---|---|---|---|
| **Partition** | Output Labels | Output Labels | Output Labels | Output Labels | Input Features | Output Labels | Output Distribution | Output Labels |
| **Model** | LSTM | LSTM | ResNet18 | ResNet18 | YOLOv8n | BiLSTM | MLP | ResNet18 |

models, (4) FL hyperparameters, and (5) IoT-factor emulator. In this section, we describe these components in detail.

### 3.2.1 Non-IID Data Partitioning

A key characteristic of FL is that data distribution at different clients is non-IID. The objective of non-IID data partitioning is to partition the training set such that data allocated to different clients follow the non-IID distribution. The eight datasets included in `FedAIoT` cover three fundamental tasks: classification, regression, and object detection. As summarized in Table 3, `FedAIoT` incorporates three different non-IID data partitioning schemes designed for the three tasks respectively.

**Scheme#1: Non-IID Partition over Output Labels.** For the task of classification (WISDM-W, WISDM-P, UT-HAR, Widar, CASAS, EPIC-SOUNDS) with $C$ classes, we first generate a distribution over the classes for each client by drawing from a Dirichlet distribution (formal definition in Appendix C) with parameter $\alpha$ (Hsu et al., 2019), where lower values of $\alpha$ generate more skewed distribution whereas higher values of $\alpha$ result in more balanced class distributions. We use the same $\alpha$ to determine the number of samples each client contains. In addition, by drawing from a Dirichlet distribution with parameter $\alpha$, we create a distribution over the total number of samples, which is then used to allocate a varying number of samples to each client, where lower values of $\alpha$ lead to a few clients holding a majority of the samples whereas higher values of $\alpha$ create a more balanced distribution of samples across clients. Therefore, this approach allows us to generate non-IID data partitions such that both class distribution and the number of samples can vary across the clients.

**Scheme#2: Non-IID Partition over Input Features.** The task of object detection (VisDrone) does not have specific classes. In such case, we use the input features to create non-IID partitions. Specifically, similar to He et al. (2021b), we first used ImageNet (Russakovsky et al., 2015) features generated from the VGG19 model (Liu and Deng, 2015), which encapsulate visual information required for subsequent analysis. With these ImageNet features as inputs, we performed clustering in the feature space using $K$-nearest neighbors to partition the dataset into clusters. Each cluster is a pseudo-class, representing images sharing common visual characteristics as per the extracted ImageNet features. Lastly, Dirichlet allocation was applied on top of the pseudo-classes to create the non-IID distribution across different clients.

**Scheme#3: Non-IID Partition over Output Distribution.** For the regression task (AEP) where output is characterized as a continuous variable, we utilize Quantile Binning (Pyle, 1999). Specifically, we divide the range of the output variable into equal groups or quantiles, ensuring that each bin accommodates roughly the same number of samples. Each

category or bin is treated as a pseudo-class. Note that the number of quantiles can be set to any value in our framework. We used ten as an example to demonstrate the results. Lastly, we apply Dirichlet allocation to generate the non-IID distribution of data across the clients.

### 3.2.2 IoT-specific Data Preprocessing

The eight datasets included in `FedAIoT` cover diverse IoT data modalities such as wireless signals, drone images, and smart home sensor data. `FedAIoT` incorporates a suite of IoT-specific data preprocessing techniques that are designed for such IoT data modalities accordingly.

**WISDM:** We followed the standard preprocessing techniques used in accelerometer and gyroscope-based activity recognition for WISDM (Ravi et al., 2005; Reyes-Ortiz et al., 2016; Ronao and Cho, 2016). Specifically, for each 3-minute session, we used a 10-second sliding window with 50% overlap to extract samples from the raw accelerometer and gyroscope data sequences. We then normalize each dimension of the extracted samples by removing the mean and scaling to unit variance.

**UT-HAR:** We followed Yang et al. (2023) and applied a sliding window of 250 packets with 50% overlap to extract samples from the raw Wi-Fi data from all three antennas. We then normalize each dimension of the extracted samples by removing the mean and scaling to unit variance.

**Widar:** We first adopted the body velocity profile (BVP) processing technique as outlined in Yang et al. (2023); Zheng et al. (2019) to handle environmental variations from the data. We then applied standard scalar normalization to normalize the data. This creates data samples with the shape of $22 \times 20 \times 20$ reflecting time axis, $x$, and $y$ velocity features respectively.

**VisDrone:** We first normalized the pixel values of drone images to range from 0 to 1. Data augmentation techniques including random shifts in Hue, Saturation, and Value color space, image compression, shearing transformations, scaling transformations, horizontal and vertical flipping, and MixUp were applied to increase the diversity of the dataset.

**CASAS:** We followed Liciotti et al. (2019) to transform the sensor readings into categorical sequences, creating a form of semantic encoding. Every possible temperature setting is assigned a distinct categorical value. Individual instances of motion and door sensor activation (on or off) are also assigned a categorical value. Subsequently, for a sensor activation, we extract a sequence consisting of the previous $2,000$ sensor activations which we then use for activity prediction.

**AEP:** Temperature data were log-transformed for skewness, and the "visibility" value indicating meteorological visibility in kilometers was binarized with median threshold. Outliers below the 10th or above the 90th percentile were replaced with corresponding percentile values. Central tendency and date features were added for time-related patterns. Principal component analysis was used for data reduction, and the output was normalized using a standard scaler.

**EPIC-SOUNDS:** We first performed a Short-Time Fourier Transform (STFT) on raw audio data followed by applying a Hanning window of 10ms duration and a step size of 5ms to ensure appropriate spectral resolution. We then extracted 128 Mel spectrogram features. To

Table 4: Representative IoT devices for each dataset.

| Dataset | Application | IoT Platform | Representative Devices | Hardware RAM Size |
|---|---|---|---|---|
| WISDM-W | | Smartwatch | Apple Watch 8 | 512 MB to 1 GB |
| WISDM-P | Activity Recognition | Smartphone | iPhone 14 | 6 GB |
| UT-HAR | | Wi-Fi Router | TP-Link AX1800 | 64 MB to 1 GB |
| Widar | Gesture Recognition | Wi-Fi Router | TP-Link AX1800 | 64 MB to 1 GB |
| CASAS | Independent Living | Smart Home | Raspberry Pi 4 | 1 GB to 8 GB |
| AEP | Energy Prediction | Smart Home | Raspberry Pi 4 | 1 GB to 8 GB |
| VisDrone | Objective Detection | Drone | Dji Mavic 3 + Raspberry Pi 4 | 1 GB to 8 GB |
| EPIC-SOUNDS | Augmented Reality | Head-mounted Device | GoPro / AR Headset | 1 GB to 8 GB |

further enhance the data, we applied a natural logarithm scaling to the Mel spectrogram output. Lastly, we padded each segment to reach a consistent length of 400.

### 3.2.3 IoT-friendly Models

Given that `FedAIoT` focuses on resource-constrained IoT devices, our choice of models is informed by a combination of model accuracies and model efficiency (Reinbothe, 2023; Yang et al., 2023; Terven et al., 2023; Liciotti et al., 2019; Seyedzadeh et al., 2018; Sholahudin et al., 2016; Huh et al., 2023). For each included dataset, we evaluated multiple model candidates, and selected the best-performing ones that adhere to the resource constraints of representative IoT devices listed in Table 4. As an example, for UT-HAR dataset, two model candidates (ViT and ResNet18) have similar accuracy, and we selected ResNet18 in our benchmark as it has better model efficiency. Table 3 lists the selected model for each dataset. The detail of the architecture of each model is described in Appendix D.

### 3.2.4 FL Hyperparameters

**Data Heterogeneity Level.** Data heterogeneity (i.e., non-IIDness) is a fundamental challenge in FL. As outlined in Section 3.2.1, `FedAIoT` facilitates the creation of diverse non-IID data partitions, which enables the simulation of different data heterogeneity levels to meet experiment requirements.

**FL Optimizer.** `FedAIoT` supports a handful of commonly used FL optimizers. In the experiment section, we showcase the benchmark results of two of the most commonly used FL optimizers: FedAvg (McMahan et al., 2017) and FedOPT (Reddi et al., 2021).

**Client Sampling Ratio.** Client sampling ratio denotes the proportion of clients selected for local training in each round of FL. This hyperparameter plays a crucial role as it directly influences the computation and communication costs associated with FL. `FedAIoT` facilitates the creation of diverse client sampling ratios and the evaluation of its impact on both model performance and convergence speed during federated training.

Details on other hyperparameters used for each experiment are described in Appendix E.

### 3.2.5 IoT-Factor Emulator

**Noisy Labels.** In real-world scenarios, IoT sensor data can be mistakenly labeled. These labels introduce noise into the federated training process. Therefore, FL systems need to be designed with robustness to label noise as a crucial feature. Label noise is a well-discussed concept in centralized training with established methods to mitigate it (Song et al., 2023). In general, label noise can be emulated in two ways: *Symmetric Noise* where the injected noise is uniformly distributed across the labels (Tanaka et al., 2018); and *Asymmetric Noise* injects noise non-uniformly across different labels using a label transition probability matrix (Patrini et al., 2017; Tanaka et al., 2018). In the context of FL, Xu et al. (2022); Zhang et al. (2023); Feng et al. (2023) propose to use symmetric noise to simulate label noise, evaluating FL algorithms at various noise levels. Fang and Ye (2022); Kim et al. (2022); Wu et al. (2023); Liang et al. (2024); Tsouvalas et al. (2024), on the other hand, propose to use asymmetric noise to simulate label noise. In particular, Fang and Ye (2022); Kim et al. (2022); Liang et al. (2024); Tsouvalas et al. (2024) design label transition probabilities in a pairwise manner, assuming that one label can only be mislabeled as another specific label with a random probability. In contrast, in `FedAIoT`, we propose a new label transition probability matrix that breaks this assumption, where a label can be mistakenly labeled as any other label with a learned probability. Specifically, we augment the ground truth labels of a dataset with a label transition probability matrix $Q$ where, $Q_{ij}$ is the probability that the true label $i$ is changed to a different label $j$, i.e., $P(\hat{y} = j \mid y = i)$. The label transition probability matrix was constructed based on centralized training results, where the elements of $Q_{ij}$ was determined from the ratio of samples labeled as $j$ by a centrally trained model to those with ground truth label $i$.

**Quantized Training.** IoT devices are resource-constrained. Previous benchmarks such as FLUTE (Garcia et al., 2022) examined the performance of quantized training during server-side model aggregation to reduce the communication cost of FL. Ozkara et al. (2021) and Gupta et al. (2023) study quantization-aware training but still use full precision during training. In contrast, `FedAIoT` incorporates not only quantized model aggregation at the server side but also quantized training at the client side to reduce the memory demands of FL on client IoT devices. Quantized training on both server and client sides is key to enabling FL for AIoT as IoT devices are not just restrained in communication bandwidth but also in on-device memory.

## 4 Benchmark Results and Analysis

We implemented `FedAIoT` using PyTorch (Paszke et al., 2019) and Ray (Moritz et al., 2018) and conducted our experiments on NVIDIA A6000 GPUs. We run each of our experiments using three random seeds and report the mean and standard deviation.

### 4.1 Overall Performance

First, we benchmark the FL performance under two FL optimizers, FedAvg and FedOPT, under low ($\alpha = 0.5$) and high ($\alpha = 0.1$) data heterogeneity levels, and compare it against centralized training.

Table 5: Overall performance.

| Dataset | Metric | Centralized | Low Data Heterogeneity ($\alpha = 0.5$) | | High Data Heterogeneity ($\alpha = 0.1$) | |
|---|---|---|---|---|---|---|
| | | | FedAvg | FedOPT | FedAvg | FedOPT |
| WISDM-W | Accuracy (%) | $74.05 \pm 2.47$ | $70.03 \pm 0.13$ | $71.50 \pm 1.52$ | $68.51 \pm 2.21$ | $65.76 \pm 2.42$ |
| WISDM-P | Accuracy (%) | $36.88 \pm 1.08$ | $36.21 \pm 0.19$ | $34.32 \pm 0.84$ | $34.28 \pm 3.28$ | $32.99 \pm 0.55$ |
| UT-HAR | Accuracy (%) | $95.24 \pm 0.75$ | $94.03 \pm 0.63$ | $94.10 \pm 0.84$ | $74.24 \pm 3.87$ | $87.78 \pm 5.48$ |
| Widar | Accuracy (%) | $61.24 \pm 0.56$ | $59.21 \pm 1.79$ | $56.26 \pm 3.11$ | $54.76 \pm 0.42$ | $47.99 \pm 3.99$ |
| VisDrone | MAP-50 (%) | $34.26 \pm 1.56$ | $32.70 \pm 1.19$ | $32.21 \pm 0.28$ | $31.23 \pm 0.70$ | $31.51 \pm 2.18$ |
| CASAS | Accuracy (%) | $83.70 \pm 2.21$ | $75.93 \pm 2.82$ | $76.40 \pm 2.20$ | $74.72 \pm 1.32$ | $75.36 \pm 2.40$ |
| AEP | $R^2$ | $0.586 \pm 0.006$ | $0.502 \pm 0.024$ | $0.503 \pm 0.011$ | $0.407 \pm 0.003$ | $0.475 \pm 0.016$ |
| EPIC-SOUNDS | Accuracy (%) | $46.97 \pm 0.24$ | $45.51 \pm 1.07$ | $42.39 \pm 2.01$ | $33.02 \pm 5.62$ | $37.21 \pm 2.68$ |

**Benchmark Results:** Table 5 summarizes our results. We make three observations. (1) Data heterogeneity level and FL optimizer have different impacts on different datasets. In particular, the performance of UT-HAR, AEP, and EPIC-SOUNDS is extremely sensitive to the data heterogeneity level. In contrast, WISDM-P, CASAS, and VisDrone show limited accuracy differences under different data heterogeneity levels. (2) Under high data heterogeneity, FedAvg has a better performance compared to FedOPT in WISDM and Widar datasets and has lower deviation for all datasets except WISDM-P and EPIC-SOUNDS. The performance gap between the two FL optimizers reduces under low data heterogeneity. (3) Compared to the other datasets, CASAS, AEP, and WISDM-W have higher accuracy margins between centralized training and low data heterogeneity.

## 4.2 Impact of Client Sampling Ratio

Second, we benchmark the FL performance under two client sampling ratios: 10% and 30%. We report the maximum accuracy reached after completing 50%, 80%, and 100% of the total training rounds for both these ratios under high data heterogeneity, thereby offering empirical evidence of how the model performance and convergence rate are impacted by the client sampling ratio.

**Benchmark Results:** Table 6 summarizes our results. We make two observations. (1) When the client sampling ratio increases from 10% to 30%, the model performance increases at 50%, 80%, and 100% of the total training rounds across all the datasets. This demonstrates the importance of the client sampling ratio to the model performance. (2) However, a higher sampling ratio does not always speed up model convergence to the same extent. For example, model convergence of CASAS and VisDrone are comparable at both sampling ratios whereas it is much faster for UT-HAR and AEP.

## 4.3 Impact of Noisy Labels

Next, we examine the impact of noisy labels on the FL performance under two label error ratios: 10% and 30%, and compare these results with the control scenario that involves no label errors. Note that we only showcase this for WISDM, UT-HAR, Widar, CASAS, and EPIC-SOUNDS as these are classification tasks, and the concept of noisy labels only applies to classification tasks.

Table 6: Impact of client sampling ratio.

| Dataset | Training Rounds | Low Client Sampling Ratio (10%) | | | High Client Sampling Ratio (30%) | | |
|---|---|---|---|---|---|---|---|
| | | 50% Rounds | 80% Rounds | 100% Rounds | 50% Rounds | 80% Rounds | 100% Rounds |
| WISDM-W | 400 | $58.81 \pm 1.43$ | $63.82 \pm 1.53$ | $68.51 \pm 2.21$ | $65.57 \pm 2.10$ | $67.23 \pm 0.77$ | $69.21 \pm 1.13$ |
| WISDM-P | 400 | $29.49 \pm 3.65$ | $31.65 \pm 1.42$ | $34.28 \pm 3.28$ | $33.73 \pm 2.77$ | $34.01 \pm 2.27$ | $36.01 \pm 2.23$ |
| UT-HAR | 2000 | $61.81 \pm 7.01$ | $70.76 \pm 2.23$ | $74.24 \pm 3.87$ | $86.46 \pm 10.90$ | $90.84 \pm 4.42$ | $92.51 \pm 2.65$ |
| Widar | 1500 | $47.55 \pm 1.20$ | $50.65 \pm 0.24$ | $54.76 \pm 0.42$ | $53.93 \pm 2.90$ | $55.74 \pm 2.15$ | $57.39 \pm 3.14$ |
| VisDrone | 600 | $27.07 \pm 3.09$ | $31.05 \pm 1.55$ | $31.23 \pm 0.70$ | $30.56 \pm 2.71$ | $33.52 \pm 2.90$ | $34.85 \pm 0.83$ |
| CASAS | 400 | $71.68 \pm 1.96$ | $74.19 \pm 1.26$ | $74.72 \pm 1.32$ | $73.89 \pm 1.16$ | $74.68 \pm 1.50$ | $76.12 \pm 2.03$ |
| AEP | 3000 | $0.325 \pm 0.013$ | $0.371 \pm 0.017$ | $0.407 \pm 0.003$ | $0.502 \pm 0.006$ | $0.523 \pm 0.014$ | $0.538 \pm 0.005$ |
| EPIC-SOUNDS | 300 | $20.99 \pm 5.19$ | $25.73 \pm 1.99$ | $28.89 \pm 2.82$ | $23.70 \pm 6.25$ | $31.74 \pm 7.83$ | $35.11 \pm 1.99$ |

Table 7: Impact of noisy labels.

| Noisy Label Ratio | WISDM-W | WISDM-P | UT-HAR | Widar | CASAS | EPIC-SOUNDS |
|---|---|---|---|---|---|---|
| 0% | $68.51 \pm 2.21$ | $34.28 \pm 3.28$ | $74.24 \pm 3.87$ | $54.76 \pm 0.42$ | $74.72 \pm 1.32$ | $28.89 \pm 2.82$ |
| 10% | $50.63 \pm 4.19$ | $28.85 \pm 1.44$ | $73.75 \pm 5.67$ | $34.03 \pm 0.33$ | $65.01 \pm 2.98$ | $21.43 \pm 3.86$ |
| 30% | $47.90 \pm 3.05$ | $27.68 \pm 0.39$ | $70.55 \pm 3.27$ | $27.20 \pm 0.56$ | $63.16 \pm 1.34$ | $13.30 \pm 0.42$ |

**Benchmark Results:** Table 7 summarizes our results. We make two observations. (1) As the ratio of erroneous labels increases, the performance of the models decreases across all the datasets, and the impact of noisy labels varies across different datasets. For example, UT-HAR only experiences a little performance drop at 10% label error ratio, but its performance drops more when the label error ratio increases to 30%. (2) In contrast, WISDM, Widar, CASAS, and EPIC Sounds are very sensitive to label noise and show significant accuracy drop even at 10% label error ratio.

### 4.4 Performance on Quantized Training

Lastly, we examine the impact of quantized training on FL under half-precision (FP16)[1]. We assess model accuracy and memory usage under FP16 and compare the results to those from the full-precision (FP32) models. Memory usage is measured by analyzing the GPU memory usage of a model when trained with the same batch size under a centralized setting. Note that we use memory usage as the metric since it is a relatively consistent and hardware-independent metric. In contrast, other metrics such as computation speed and energy are highly hardware-dependent. Depending on the chipset that the IoT devices use, the computation speed and energy could exhibit wide variations. More importantly, new and more advanced chipsets are produced every year. The updates of the chipsets would inevitably make the benchmarking results quickly obsolete and out of date.

**Benchmark Results:** Table 8 summarizes the model performance and memory usage at two precision levels. We make three observations: (1) As expected, the memory usage significantly decreases when using FP16 precision, ranging from 57.0% to 63.3% reduction across different datasets. (2) Similar to Micikevicius et al. (2018), we also observe that model performance associated with the precision levels varies depending on the dataset. For AEP

---

1. PyTorch does not support lower quantization levels like INT8 and INT4 during training as referenced Pytorch (2023) at the time of writing and hence those were excluded. Additionally, FP8 is only supported for H100 GPUs, not the mobile GPUs used for IoT devices.

Table 8: Performance on quantized training.

| Dataset | Metric | FP32 | | FP16 | |
|---------|--------|------|--|------|--|
| | | Model Performance | Memory Usage | Model Performance | Memory Usage |
| WISDM-W | Accuracy (%) | $68.51 \pm 2.21$ | 1444 MB | $60.31 \pm 5.38$ | 564 MB ($\downarrow 60.9\%$) |
| WISDM-P | Accuracy (%) | $34.28 \pm 3.28$ | 1444 MB | $30.22 \pm 2.05$ | 564 MB ($\downarrow 60.9\%$) |
| UT-HAR | Accuracy (%) | $74.24 \pm 3.87$ | 1716 MB | $72.86 \pm 4.49$ | 639 MB ($\downarrow 62.8\%$) |
| Widar | Accuracy (%) | $54.76 \pm 0.42$ | 1734 MB | $34.03 \pm 0.33$ | 636 MB ($\downarrow 63.3\%$) |
| VisDrone | MAP-50 (%) | $31.23 \pm 0.70$ | 8369 MB | $29.17 \pm 4.70$ | 3515 MB ($\downarrow 60.0\%$) |
| CASAS | Accuracy (%) | $74.72 \pm 1.32$ | 1834 MB | $72.86 \pm 4.49$ | 732 MB ($\downarrow 60.1\%$) |
| AEP | $R^2$ | $0.407 \pm 0.003$ | 1201 MB | $0.469 \pm 0.044$ | 500 MB ($\downarrow 58.4\%$) |
| EPIC-SOUNDS | Accuracy (%) | $33.02 \pm 5.62$ | 2176 MB | $35.43 \pm 6.61$ | 936 MB ($\downarrow 57.0\%$) |

and EPIC-SOUNDS, the FP16 models improve the performance compared to the FP32 models. (3) Widar and WISDM-W have a significant decline in performance when quantized to FP16 precision.

### 4.5 Insights from Benchmark Results

**Need for Resilience on High Data Heterogeneity:** As shown in Table 5, datasets can be sensitive to data heterogeneity. We observe that UT-HAR, Widar, AEP, and EPIC-SOUNDS show a significant impact under high data heterogeneity. These findings emphasize the need for developing advanced FL algorithms for data modalities that are sensitive to high data heterogeneity. Handling high data heterogeneity is still an open question in FL (Zhao et al., 2018; Li et al., 2020a) and our benchmark shows that it is a limiting factor for some of the IoT data modalities as well. To mitigate high data heterogeneity, although many techniques have been proposed (Sattler et al., 2019; Arivazhagan et al., 2019), their performance is still constrained. Most recently, some exploration on incorporating generative pre-trained transformers (GPT) as part of the FL framework has shown great performance in mitigating high data heterogeneity (Zhang et al., 2024).

**Need for Balancing between Client Sampling Ratio and Resource Consumption of IoT Devices:** Table 6 reveals that a higher sampling ratio can lead to improved performance in the long run. However, higher client sampling ratios generally entail increased communication overheads and energy consumption, which may not be desirable for IoT devices. Therefore, it is crucial to identify a balance between the client sampling ratio and resource consumption.

**Need for Resilience on Noisy Labels:** As shown in Table 7, certain datasets exhibit high sensitivity to label errors, significantly deterring FL performance. Notably, both WISDM-W and Widar experience a drastic decrease in accuracy when faced with a 10% label noise ratio. Given the inevitability of noise in real FL deployments where private data is unmonitored except by the respective data owners, the development of label noise resilient techniques becomes crucial for achieving reliable FL performance. However, handling noisy labels is a much less explored topic in FL. Pioneering work on this topic is still exploring the effects of noisy labels on the performance of FL (Zhang et al., 2023; Feng et al., 2023). To mitigate the effect of noisy labels, for low label noise rates, techniques such as data augmentation and regularization have been shown to be effective in centralized training settings (Shorten and

Khoshgoftaar, 2019; Zhang et al., 2018). If the label error rate is high, techniques such as knowledge distillation (Hinton et al., 2015), mixup (Zhang et al., 2018), and bootstrapping (Reed et al., 2015) have been proposed. We expect such techniques or their variants would help in the FL setting.

**Need for Quantized Training:** Table 4 highlights the need for quantized training given the limited RAM resources on representative IoT devices. From our results summarized in Table 8, we observe that FP16 quantization does not affect accuracy drastically unless normalization layers are present, like batch normalization (Jacob et al., 2018). To mitigate this issue, techniques such as incremental quantization (Zhou et al., 2017) or ternary weight networks (Liu et al., 2023) have shown great performance in the centralized training setting. We expect such techniques or their variants would help in the FL setting as well.

## 5 Conclusion

In this paper, we presented `FedAIoT`, an FL benchmark for AIoT. `FedAIoT` includes eight datasets collected from a wide range of authentic IoT devices as well as a unified end-to-end FL framework for AIoT that covers the complete FL-for-AIoT pipeline. We have benchmarked the performance of the datasets and provided insights on the opportunities and challenges of FL for AIoT. Moving forward, we aim to foster community collaboration by launching an open-source repository for this benchmark, continually enriching it with more datasets, algorithms, and thorough analytical evaluations, ensuring it remains a dynamic and invaluable resource for FL for AIoT research.

## 6 Acknowledgement

We would like to thank the action editor Peter Mattson and the anonymous reviewers of the Journal of Data-centric Machine Learning Research for their helpful and constructive comments. Zhichao Cao is supported in part by National Science Foundation (NSF) under award NeTS-2312675. JeongGil Ko is partially supported by the Ministry of Science and ICT (MSIT) and the IITP (IITP-2022-2020-001461, IITP-2022-0-00420) in the Republic of Korea. Tiantian Feng and Shrikanth S. Narayanan are supported in part by USC + Amazon Center on Secure & Trusted ML. Tuo Zhang and Salman Avestimehr are supported in part by ARO award W911NF1810400, ONR Award No. N00014-16-1-2189 and NSF under NSF SATC ProperData Center. Samiul Alam and Mi Zhang are supported in part by the Meta Reality Labs Faculty Research Award. The views, opinions, and/or findings expressed are those of the author(s) and should not be interpreted as representing the official views or policies of the Department of Defense or the U.S. Government.

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
