# OpenReview forum: "FedAIoT: A Federated Learning Benchmark for Artificial Intelligence of Things"
_DMLR — Accepted by DMLR_

### Review · Reviewer_itEF · 2024-03-21

**Recommendation:** 3
**Confidence:** 2

**Summary Of Contributions:**

Thanks for submitting to DMLR! This paper introduces a benchmark for FL in the realm of IoT. This benchmark is built on top of eight existing datasets collected from IoT devices and are targeting representative applications of IoT. This type of benchmark is aligned with the theme of data-centric machine learning. I also agree there is a gap between federated learning on IoT devices and the datasets being used (although I am not an expert in FL).

The benchmark suite and dataset are made open source and available under a permissive licence.

Overall I like the idea of FL benchmarks and I think the theme is well aligned with the DMLR journal. However, I also have some questions/suggestions.

**Strengths:**

- I particularly appreciate the authors' effort in preparing the diverse sizes of the dataset, such that computing-poor could still benefit it. It would be more interesting too explore how different algorithms scale given different computing resources.
- The authors clearly described the datasets they used to build the benchmark: with a brief overview of the eight datasets, how data is partitioned, preprocessing and how noisy labels are introduced. I personally enjoyed reading the description and the process looks plausible to me (basically standard approach or following previous work).
- I like the idea of introducing quantized training into the context of FL. It might be a standard approach but it's great to see it being part of a benchmark. I have some questions/suggestions about quantized training, though (see below).

**Audience:**

Yes

**Broader Impact Concerns:**

I don't have any concerns.

**Claims And Evidence:**

Yes, as far as I can tell the claims are supported by evidence.

**Datasets And Benchmarks:**

It is a benchmark including the link to the open source repository and documentation. It is permissively licensed under Apache-2. There're documentation for reproducibility.

**Extended Submissions:**

No

**Limitations:**

see "requested changes" :)

**Requested Changes:**

- On page 2 you mentioned "...These datasets, however, do not originate from authentic IoT devices..." and exemplified it with CIFAR-10 and CIFAR 100 - but at the same time you also described some datasets in the FL realm (e.g., in table 2). Are you suggesting that even though there exist some datasets, but most FL works do not use them? If that is the case - I agree there's a gap - but more on the datasets side. Anyway, It would be great if you could actually list some recent FL works and analyse what datasets they are using. For readers (e.g., me) who are not very familiar with recent FL work it is interesting to see this info, and it helps strengthen the motivation of this work.
- I like the idea of introducing quantized training. Though you specifically mentioned "The updates of the chipsets would inevitably make the benchmarking results quickly obsolete and out of date." - which I agree. But as an outsider, I still felt it is useful to have some sort of benchmarking results in terms of training throughput and memory usage. It could serve as a weak baseline.
- Another minor question regarding quantized training - is anyone in the FL regime working on even lower precision training? I know there's fp8 training in LLMs, but unsure about FL - just curious.
- Noisy Labels: "In real-world scenarios, IoT sensor data can be mistakenly labeled.". Again I am not FL person, so I am quite curious about the flow. When you say mistakenly labeled, who labeled them? I assume this is only for the benchmarking dataset, but realistically, since those data are collected by IoT sensors and in an FL environment, there's no "central annotators", how's those data labeled before training? I am asking because I feel that, knowing how those errors are introduced might be helpful in synthesise more realistic noisy labels. The [dataperf paper](https://arxiv.org/abs/2207.10062) might be interesting as it also includes some noisy label generation.
- I would suggest another round of proofread. For example, in the abstract, "in this work, we introduce FedAIoT, an FL benchmark for AIoT to fill this critical gap.". I think there should be a comma between "an FL benchmark for AIoT" and "to fill this...".

**Strengths And Weaknesses:**

- Good relevance to the DMLR community.
- Dataset are well chosen and facilitates researchers with different computing power.
- The paper is well-written and easy to follow.
- I like the idea of introducing quantized training - which is also very relevant to FL community as far as I am aware.

---

### Review · Reviewer_1xQT · 2024-04-26

**Recommendation:** 4
**Confidence:** 3

**Summary Of Contributions:**

The authors introduce a new federated learning benchmark for AI of Things (AIoT). The new benchmark consists of eight datasets collected from IoT devices. The benchmark results are collected from a wide range of experimental setups, including various levels of client sampling ratio, noisy label ratio, data heteregenoity, and quantized training.

**Strengths:**

See above.

**Audience:**

Yes

**Broader Impact Concerns:**

It is not entirely clear to me how the data is collected from IoT devices. The authors discuss the "methods" used for measurement, but I think more information on the measurement conditions and their diversity would be useful. For instance, for EPIC-SOUNDS, how was the raw audio data selected?

**Claims And Evidence:**

Yes

**Datasets And Benchmarks:**

I think the data collection procedure could be more detailed as suggested above.

**Extended Submissions:**

NA

**Limitations:**

I don't see a major limitation.

**Requested Changes:**

A brief explanation on the heterogeneity comment above would be great.

**Strengths And Weaknesses:**

## Strengths:

- The authors fill a gap in the federated learning research by introducing this new benchmark specifically for IoT devices.

- This benchmark, together with its extensive analysis of various settings, has the potential to provide a tool for comparison between different federated learning strategies.

- The authors cover a range of realistic scenarios varying the level of client sampling ratio, noisy label ratio, data heterogeneity, and quantized training.


## Weaknesses:

- The authors propose various ways to make the collected data heterogeneous, and all of them make sense. But, since the data is already collected from real-world IoT devices, doesn't it already have some natural heterogeneity that reflects the real world more realistic than a Dirichlet allocation? I couldn't follow why the collected data itself is not considered as a heterogeneous dataset or why the proposed ways would be better than the initial dataset collected from different IoT devices.

---

### Review · Reviewer_yJ15 · 2024-06-22

**Recommendation:** 3
**Confidence:** 3

**Summary Of Contributions:**

The paper introduces FedAIoT, a benchmark for FL specifically focused on Intelligence of Things applications. The benchmark comprises eight datasets collected from various real-world IoT devices and including modalities like accelerometer data, wireless signals, drone images, and smart home sensor readings. It also includes a FL framework designed for IoT, encompassing non-IID data partitioning, IoT-specific preprocessing, and emulators for noisy labels and quantized training.

**Strengths:**

See above.

**Audience:**

Yes

**Broader Impact Concerns:**

- The paper does not explicitly address potential biases present in the collected datasets.
- The benchmark's focus on IoT applications raises considerations regarding data privacy, security, and potential misuse of collected information. FL by itself does not provide any guarantee regarding privacy.
- Human activity recognition with Wi-Fi router seems like a perfect use case for mass surveillance. What was the key motivation to include it? Where it is really used in real-world setting?

**Claims And Evidence:**

Somewhat covered.

**Datasets And Benchmarks:**

It can be improved. See my comments above.

**Extended Submissions:**

N/A

**Limitations:**

- While eight datasets offer a good starting point, expanding the benchmark with additional datasets covering a wider range of IoT applications would further enhance its value. The choice of the considered datasets is rather weak.
- The current benchmark emulates resource constraints but could benefit from incorporating more realistic device heterogeneity in terms of computational power, communication bandwidth, and availability.
- Lack of comparison with prior approaches and missing references.

**Requested Changes:**

- The current benchmark primarily focuses on FedAvg and FedOPT, with unclear justification of these strategies. Including additional SOTA FL algorithms (e.g., FedProx, SCAFFOLD) would provide a more comprehensive evaluation of different approaches.
- Provide clarification on dataset licensing as some datasets mention licenses, others don't.
- What is the impact of model architecture?

**Strengths And Weaknesses:**

+ The benchmark seems to fill a gap by focusing exclusively on FL for IoT and capturing unique challenges inherent to this domain.
+ It includes diverse datasets and modalities.
+ The framework can simplify benchmarking in FL as it offers a standardized pipeline for data partitioning, model training, and evaluation.
- The choice of the datasets seems largely arbitrary and rationale for their selection is weak.
- There is already a significant work done in FL for sensors and learning from noisy labels, how this work is any different is unclear.
- The IMU related datasets are not really representative and small in size, there are other bigger datasets more suited for such tasks like ExtraSensory.
- Learning from noisy label section is quite weak and no comparisons are provided with prior work and paper simply propose its own way of creating noisy dataset, ignoring several other important works. I suggest authors to do their own research to find relevant references.
- Quantized training in FL setting is also not new and explored earlier extensively along with knowledge distillation methods.